# Characterization of Forest Ecosystems in the Chure (Siwalik Hills) Landscape of Nepal Himalaya and Their Conservation Need

**Yadav Uprety [1],\*, Achyut Tiwari [1], Sangram Karki [2], Anil Chaudhary [1], Ram Kailash Prasad Yadav [1], Sushma Giri [3], Srijana Shrestha [3], Kiran Poudel [3] and Maheshwar Dhakal [3],\***

[1] Central Department of Botany, Tribhuvan University, Kathmandu 44600, Nepal
[2] Forest Research and Training Center, Ministry of Forests and Environment, Babarmahal, Kathmandu 44600, Nepal
[3] President Chure Terai-Madhesh Conservation Development Board, Khumaltar, Lalitpur 44700, Nepal
\* Correspondence: yadavuprety@gmail.com (Y.U.); maheshwar.dhakal@gmail.com (M.D.)

**Abstract:** As a basic component of the forest ecosystem, the forest structure refers to the general distribution of plant species of different life forms and sizes. The characterization of forest structure is the key to understanding the vegetation history, present status, and future development trajectory of the forest ecosystems. The Chure region of Nepal covers about 12.78% of the country's land area and extends east to west along the southern foothills. This biologically rich but geologically fragile region is home to many species and provides many ecosystem services to millions of people. The Chure landscape is severely suffered from anthropogenic disturbances including logging, grazing, fuelwood collection, solid waste disposal, encroachment, forest fire, and excavation of sand, gravel, and boulders. In this study, we aim to characterize the forest ecosystem types outside the protected areas in the Chure region of Nepal and analyze the threat and vulnerability of the landscape from the biodiversity point of view. We sampled 62 sites to study the dominant vegetation type, regeneration status, and major threats to the forest ecosystems. A distribution map of the forest ecosystem types in Chure was prepared. We identified 14 forest ecosystem types in Chure including seven new ones. The newly reported forest ecosystems are *Hymenodictyon excelsum* Forest, *Syzygium cumini* Forest, *Terminalia anogeissiana* Forest, *Schima wallichii–Shorea robusta* Forest, *Pinus roxburghii–Shorea robusta* Forest, *Pinus roxburghii* Forest, and Bamboo thickets. We conclude that intensified human activities including forest encroachment and deforestation are mainly responsible for the ecological imbalance in the Chure region. We emphasize an in-depth analysis of biophysical linkage and immediate conservation efforts for the restoration of the Chure landscape in Nepal.

**Keywords:** biodiversity; disturbances; ecosystem; forest types; threats

## 1. Introduction

Forest structure is both a product and a key driver of biological diversity and ecosystem processes over a long temporal and spatial scale [1–3]. Forest structure affects forest productivity, tree species diversity, and biological habitat that eventually determines the quality of forest ecosystem goods and services [1,2,4–6]. A particular forest stand structure is a basic unit of a forest community and also a proxy for a specific biological community. Such a community with a dominance of particular species and homogenous environmental parameters with relatively stable conditions forms a unique habitat and represents an ecological facet that evolves as a unique forest ecosystem [7,8]. A local-scale forest ecosystem is characterized by spatially co-occurring vegetation assemblages that share a common ecological gradient, substrate, or process [9]. The interconnections among local-scale ecosystems in different spatial and temporal scales form a hierarchical structure and

shape the nature of future ecosystems [10]. The precise spatiotemporal information on forest types and areas at the regional scale is required for their better management, understanding of the carbon cycle, and modeling of biophysical attributes, hydrology, and climate [6,11]. Understanding threats and vulnerabilities to forest biodiversity is needed for actions to slow the current risks and secure ecosystem services for future generation [12].

The Chure mountain range (also called Siwalik hills) forms a more than 2000 km stripe along the outer Himalayas through India to Nepal and into northern Pakistan [13]. Situated between the plains in the south and the Mahabharat hills in the north, the Chure is one of the youngest mountains in the world [14]. The Nepal part of the Chure spreads over 37 districts and covers 12.78% of the country's total land area, forming a stretch of 800 km east to west [15–19]. Besides the lower plains of the Indian subcontinent, Chure also constitutes the larger parts of inner valleys (also called Dun valleys) in Udayapur, Sindhuli, Makawanpur, Chitwan, Surkhet, and Dang districts in Nepal [18]. The average peak is high in west Nepal at about 1800 m and low in east Nepal with a maximum of 700 m altitude [20]. It is the most fragile and vulnerable ecosystem because of natural and anthropogenic factors [14,16,17,21]. Due to the weak and fragile geography, the Chure region is highly vulnerable to erosion and other hazards. For these reasons, coupled with poor water availability, Chure was not inhabited in the past. However, with increasing population pressure from migration after the 1980s the areas of cultivation had begun to appear in these hills which was regarded as a very undesirable trend in view of the extremely fragile nature of the soil in this area [20].

The Chure region is severely affected by various anthropogenic activities. The people living in core Chure forest areas are heavily dependent on forest resources [22]. Along with population growth and migration, the region is threatened by deforestation, grazing, fuelwood collection, encroachment, forest fire, and excavation of sand, gravel, and boulders. Illegal logging and excavation of sand, gravel, and boulders and haphazard development activities without consideration of environmental impacts are mostly responsible for the degradation of the Chure landscape. The environmental integrity of the region has been severely altered because of these drivers of change. Realizing the ecological importance and conservation sensitivity and needs, the Government of Nepal formulated the 20-year Master Plan for sustainable conservation of the region [17]. Chure conservation is one of the national priority programs of the Government of Nepal for the last two decades [23].

A number of studies provide important information about conservation and management issues [24–26], biodiversity, and ecosystem services [16,19,22,27–29], landscape processes [14,30], and agroforestry systems [31] in Chure. The Master Plan has mentioned 11 forest ecosystems outside protected areas (eight in Chure and three in Terai regions) referring to Biodiversity Profile Project (BPP), but the location and conservation status of these ecosystems are unknown [17]. The BPP conducted during 1994–1996 was largely based on the ecological maps produced during 1971–1985 and the reports have highlighted the potential inconsistencies in these ecological maps [15,32]. Considering the basic forest structure as a proxy of specific forest ecosystem type, we aim to identify the forest ecosystems outside protected areas, locate these ecosystems in the map, assess their status, and identify important biodiversity areas in Chure landscape. Moreover, we also present the important knowledge synthesis about the major biological features of the Chure. Characterizing these ecosystems would help deploy proper conservation and management strategies by providing tools for communicating the relevance of ecosystems to the public, and support decision-makers to spatially identify priority areas.

## 2. Materials and Methods

### 2.1. Study Area

The study covers the Chure landscape of the Nepal Himalaya (Figure 1). Chure represents the parts of Nepal's tropical and sub-tropical bioclimatic regions and forms the largest and longest landscape. Of the total forest area of Nepal, 23.04% lies in Churia [33]. Chure covers parts of all the seven provinces of Nepal and also the parts of the Terai Arc Landscape (TAL), Chitwan-Annapurna Landscape (CHAL), and Kangchenjunga Landscape (KL) [34–36]. About 7.7 million (26% of the national population) people live in 37 districts of Chure [37].

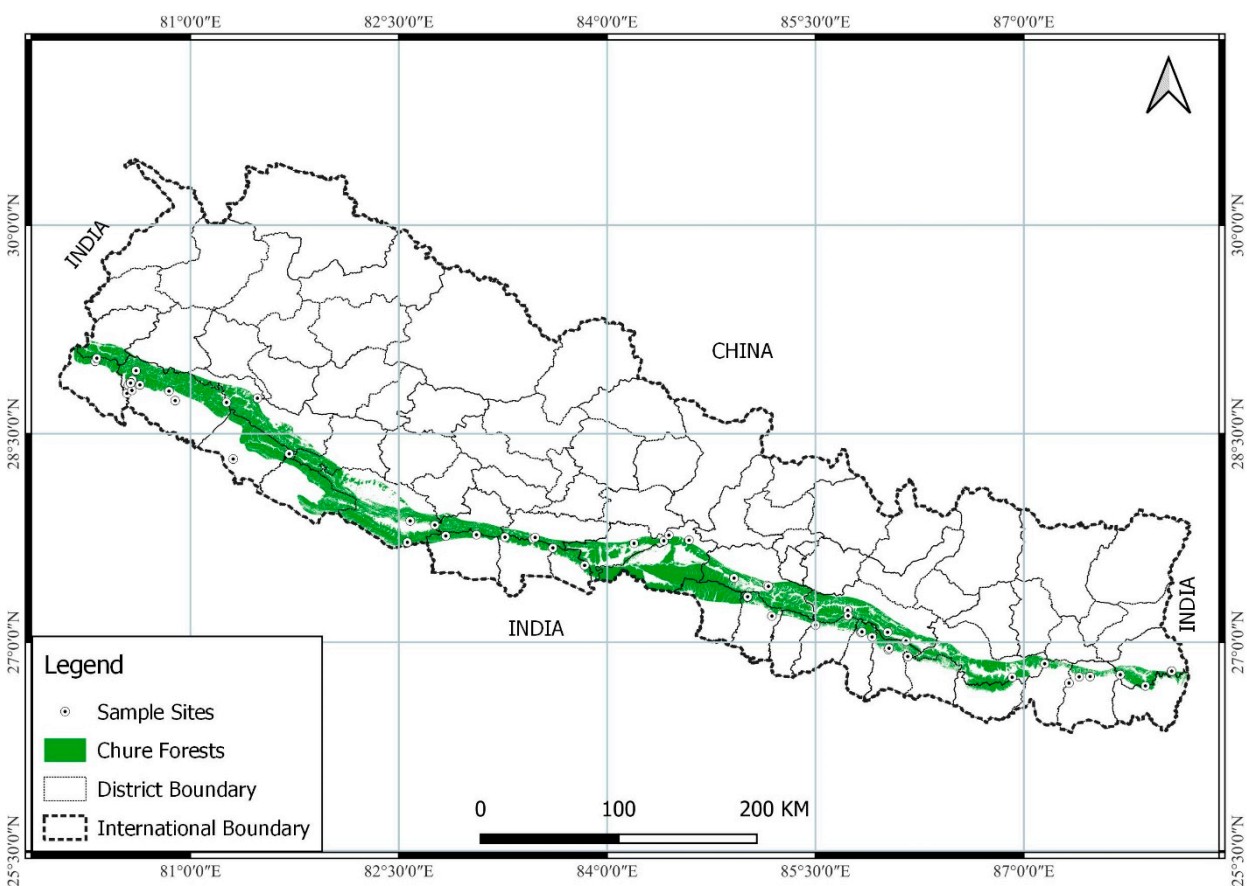

**Figure 1.** Map of Nepal showing Chure range and forests in green. The black dots represent the sampling sites.

The Chure are formed from soft, very erodible sediments so that gullies and areas of bad-land are frequent. It is believed that Chure hills have been formed from sediments produced by the rising Himalaya during the last 40 million years or so. Chure is geologically young and composed of unconsolidated loose materials originating from soft rocks (e.g., mudstone, sandstone, siltstone, and shale) [13,18].

### 2.2. Methods

Primary data were collected from the field and secondary information was generated from the various studies including that of Dobremez [38], BPP reports published during 1995–1996 [15,32], and others [16,17,24,25,27,29,39]. Moreover, participatory and consultative approaches were also adopted to identify the sampling sites and threats to the forest ecosystems.

2.2.1. Sampling Procedures

We located forest ecosystems outside the protected areas in the Chure landscape based on the literature [15], satellite images, and expert knowledge. Forests were identified both by the presence of trees (5 m height) and the absence of other predominant land uses following FAO (2000) (land with a tree canopy cover of more than 10 percent and an area of more than 0.5 ha) [8]. Based on this working definition, a forest spreading over 0.5 ha was considered as a distinct forest ecosystem. The forest with more than 60% dominance was named after that species and the forest with the dominance of more than one species was named after those species, for example *Schima wallichii–Shorea robusta* Forest, following the approach used by FRTC [40].

We adopted random-purposive sampling for gathering primary information on forest ecosystems, species composition, and conservation status. A total of 62 sampling sites covering 24 districts were studied (Figure 1). At each sampling site (within the 0.5 ha forest), a plot size of 20 m × 20 m was established. Dominant and co-dominant tree species and their coverage, and other associated species including shrubs and herbs were noted. The sampled forest site located at the lowest elevation was at 134 m asl (Baba Tal, Sarlahi) while the plot at 1355 m asl (Khanidanda Chure Rural Municipality at Kailali district) represents the highest elevation site. Coordinates of each plot were noted using the Global Positioning System and later the distribution map of the forest ecosystem types was prepared. Field visits were conducted from November 2021 to February 2022.

2.2.2. Assessment of Regeneration Status and Disturbances

The regeneration of the forest species and disturbances to the forest ecosystems were studied inside the 20 m × 20 m plots. Each plot was subdivided into 10 m × 10 m and the regeneration status (number of seedlings and saplings) was noted in the clockwise direction within the plots. The regeneration status of the forest is expressed with categorical variables, i.e., low = 1 (less than 20 individuals of seedlings and saplings), moderate = 2 (more than 20 and less than 60 individuals), and high = 3 (more than 60 individuals). Then, the average regeneration score of each forest type was calculated. Such a mixed approach of qualitative and quantitative methods is common in forest status assessments [16,41]. Likewise, disturbances such as forest encroachment and deforestation (based on the evidence of grazing, cut stumps/logging, fuelwood collection, fodder collection), forest fire, and the presence of invasive species were noted and assigned scores from 0 to 4 (0 = no disturbance; no sign of disturbance; 1 = low disturbance; with one evidence of disturbance; 2 = moderate disturbance; with two evidences; 3 = high disturbance; with three evidences, and 4 = very high disturbance; with more than three evidences). Based on the score obtained, the threat level was identified. The disturbance indicators used for this assessment are the common factors responsible for deforestation and forest degradation in Nepal [16,42], and the scale is comparable to Miehe et al. [39].

2.2.3. Stakeholder Consultation

Participatory and consultative (stakeholder involvement) approaches were also used to locate the sampling sites and validate the information collected from the field. For this purpose, we organized seven consultative workshops one in each province. Eight to twelve participants from the communities and forest authorities participated in each consultation workshop and the participants were asked to locate the different forest types on the map. Further, they were asked to highlight the status and observed threats to these forests. The disturbance score and the threat assessment (asterisks) were validated in these consultative workshops.

2.2.4. Plant Identification

In cases where field identification was certain, for example, *Shorea robusta, Pinus roxburghii,* and *Schima wallichii,* plant species were identified on the spot. In other cases, field

notes, local names, and photographs were taken and herbarium specimens were collected. The specimens were identified with the help of reference collections [43–48] and expert consultation.

## 3. Results and Discussion

### 3.1. Forests and Ecosystems Types

Based on the dominant and co-dominant tree species type as one of the major characteristics of forest ecosystem, 14 forest ecosystems outside protected areas in Chure were identified, including the Bamboo thickets in the Siraha district of east Nepal, and located in the map (Tables 1 and S1, Figures 2 and 3). The ecosystem types presented in BPP report [15] were further validated and additional ecosystem types were identified from this study (Table 2). The most common forest ecosystem types in Chure were *Shorea robusta* (39%), followed by Tropical mixed broadleaved forest (25%) whereas *Hymenodictylon excelsum*, *Senegalia catechu*, and *Albizia* forests were poorly represented.

As our study took reference of BPP report [15] (referred also in Master Plan of Chure [17]) to locate the forest ecosystems in Chure (Table S2), we attempted to locate all forest ecosystems including *Alnus nitida* forest in the Chure region. However, our study did not find the *Alnus nitida* forest. The occurrence of this forest species was further verified with the flora of far-west Nepal [47] and experts having knowledge about the local flora of western Nepal. It was confirmed that the species is not present in Chure. This species is a component of Mugu Karnali vegetation in the Humla region at about 2100 m [49]. Hara et al. [44] have also mentioned *Alnus nitida* as a component of western flora. One of the possible reasons why this ecosystem type does not exist could be some of the inconsistencies in the ecological maps produced by Dobremez and others during 1971–1985 as reported by BPP [32].

A total of 118 ecosystems have been identified in Nepal, including 112 forest ecosystems, four cultivation ecosystems, one water body ecosystem, and one glacier/snow/rock ecosystem [15]. Among the five physiographic zones found in Nepal, the Middle Mountains have the highest number (53) of ecosystems. The High Himal and High Mountains combined have 38 ecosystems. The Terai and Siwalik have 14 and 12 ecosystems respectively [15,32]. TISC [50] reduced the 118 types to 36, excluding the Nival zone and the water bodies [51]. This information needs to be updated as natural ecosystems are dynamic in nature, and their characteristics can vary over time [52,53].

**Table 1.** Forest ecosystem types with their characteristic vegetation and general description in Chure (see Table S1 for location of the ecosystem types and GPS Coordinates).

| SN | Ecosystem Type and Threat Level [#] | Representative Location(s) in Chure | Characteristic Vegetation | Description |
|---|---|---|---|---|
| 1 | *Shorea robusta* forest (**Sal ban**) (128–1110 m asl) ** | Kanchanpur (Daiji, Bedkot-1), Kailali (Syaule), Dang (Rapti, Bhalubang), Arghakhachi (Pirapani), Palpa (Bhutkhola, Dovan), Rupandehi (Debdaha), Nawalparasi (Daunee, Debchuli, Maulakalika), Tanahun (Pipaltar), Makwanpur (Jayasingh Manohari), Bara (Niggad), Sindhuli (Fulbari Marine Gaupalika, Maddovan, Marin, Ranibas, Kalapani), Mahottari (Tuteshor), Dhanusa (Bhatighari), Udayapur (Sundarpur), Sunsari (Latijoda), Ilam (Shikharkateri). | **Trees:** *Shorea robusta, Terminalia alata, Aegle marmelos, Trewia nudiflora, Lagestroemia parviflora, Ziziphus mauritiana, Engelhardia spicata, Syzygium cumini, Mallotus philippensis* **Shrubs:** *Carissa carandas, Woodfordia fruticosa, Justicia adhatoda, Clerodendron viscosum, Cycas pectinata* | *Shorea robusta* forms the pure forest at lower elevations and predominates the flat places from east to west, at 900 m to 1100 m elevations. Sal forest associates with *Schima wallichii*, another semi-deciduous species in Central and Eastern Nepal at around 1100 m. |
| 2 | *Hymenodictyon excelsum* forest (**Latikarma ban**) (270 m asl) ** | Morang | **Trees:** *Hymenodictyon excelsum, Shorea robusta* **Shrubs:** *Murraya koenigii, Clerodendron viscosum* | Pure stand of this species is found along the flood plains of Chure in eastern Nepal. |
| 3 | *Syzygium cumini* forest (**Jamun ban**) (186–237 m asl) * | Kailali (Masuriya), Karnali flood plain | **Trees:** *Lagestroemia parviflora, Terminalia alata, Dalbergia sissoo* **Shrubs:** *Clerodendron viscosum, Colebrookea oppositifolia* | Primary evergreen forest of *Syzygium cumini* is found in western Terai. It replaces *Shorea robusta* forest along the large riversides in moist and shady areas, *Syzygium cumini* is also one of the major associated species of other forest types-*Shorea robusta* forest, *Terminalia* forest, *Terminalia anogeissiana* forest. |
| 4 | *Terminalia anogeissiana* forest | Palpa (Majhuwa, Kanchakhola) Surkhet (Pokharikada) | **Trees:** *Terminalia anogeissiana, Shorea robusta, Terminalia alata* | The species occurs from Terai to about 1700 m, usually in Sal forest but it is also |

| | | | | |
|---|---|---|---|---|
| | (**Banjhi ban**) (1073 m asl) * | | **Shrubs:** *Woodfordia fruticosa, Maesa montana* | a common constituent of rather dry forest in the Chure, particularly in western Nepal, where it is sometimes dominant and form a distinct ecosystem. |
| 5 | *Terminalia* forest (**Sajh ban**) (386–468 m asl) *** | Kanchanpur (Salghad), Kailali (Godawari) | **Trees:** *Terminalia alata, Terminalia chebula, Shorea robusta, Mallotus philippensis, Terminalia anogeissiana, Trichilla connaroides, Buchanania latifolia* **Shrubs:** *Woodfordia fruticosa, Justicia adhatoda, Colebrookea oppositifolia* | These types of forests are often mixed with tropical *Shorea robusta* forest, but in some places they form the pure stand with more than 60 % coverage. *Terminalia alata* is the predominant species of this type. |
| 6 | *Senegalia catechu* forest (**Khair ban**) (**134–185 m asl**) **** | Sarlahi (Bagmati- 11, Lopchan Tol), Dhanusha (Bhatighari CF Puspabanpur), Siraha (Baba Tal, Bandipur-3) | **Trees:** *Senegalia catechu, Albizia procera, Syzygium cumini, Mallotus philippensis, Azadirchata indica, Lagestroemia parviflora, Terminalia alata, Dalbergia latifolia, Bridelia retusa* **Shrubs:** *Murraya koenigii, Urena lobata* | Mature secondary forest of these types is very much limited due to deforestation, however in some areas of eastern Nepal, primary forest of *Senegalia catechu,* can be observed associated with *Dalbergia sissoo* along the river banks. |
| 7 | *Albizia* forest (**Siris ban**) (144–700 m asl) ** | Sunsari (Chatara) | **Trees:** *Albizia procera, Albizia lebbeck, Adina cordifolia, Cassia fistula, Alstonia scholaris, Wendlandia exserta* **Shrubs:** *Murraya koengii, Clerodendron viscosum* | *Albizia procera*, a semi-deciduous tree and *A. lebbeck* occurs in dry open forest and at the Sal forest margins in Chure-Terai region of Central and Eastern Nepal. *Albizia procera* associates with *Albizia julibrissin, Albizia chinensis* and *Erythrina stricta* in south facing slopes of outer foothills at the edge of abandoned land terrace. |
| 8 | *Dalbergia sissoo–Senegalia catechu* forest (**Sisoo–Khair ban**) (195–346 m asl) *** | Dang (Lamahi), Kailali (Godawari, Malakheti, Geta, Shreepur) | **Trees**: *Senegalia catechu, Terminalia alata, Dalbergia sissoo, Syzygium cumini, Aegle marmelos, Shorea robusta, Sapium insigne, Terminalia anogeissiana, Lagerstroemia parviflora* **Shrubs**: *Murraya koenigii, Randia spinosa, Urena lobata, Colebrookea oppositifolia* | Forest as a discrete patch close to the river edge on newly formed gravels or midstream islands created by floods from Chure rivers. Forest types of Sisoo-Khair abundance of various heights, resulting in a discontinuous canopy with poor understory. |

| | | | | |
|---|---|---|---|---|
| 9 | Tropical deciduous riverine forest (**Usna Pradeshya Nadi Tatiye Pathjhar ban**) (256–321 m asl) ** | Kailali (Godawori), Bardiya (Padanaha, Chepang), Ilam | **Trees:** *Bombax ceiba, Tetrameles nudiflora, Sapium insigne, Holopetela integrifolia, Adina cordifolia, Terminalia alata, Dalbergia sissoo, Senegalia catechu.* **Shrubs:** *Murraya koengii, Colebrookea oppositifolia* | It is found along the streams of Bhabar and Dun valleys. In west and central Nepal major component of this forest types are *Bombax ceiba*, *Adina cordifolia* and *Sapium insigne*, however in east Nepal the species composition is different, where the area is dominated by deciduous *Tetrameles nudiflora* and other associated species like *Alangium salviifolium* and *Toona ciliata*. |
| 10 | Tropical mixed broadleaved forest (**Usna Pradeshiye Misrit Chaudapate ban**) (151–1026 m asl) *** | Bara (3 no. Khola), Makawanpur (Gadhi), Morang (Thakaldada), Jhapa (Kankai), Ilam (Jorkalas), Kanchanpur (Daiji Bedkot, Libna), Kailali (Mohaniyal), Dang (Koilabash thulichaur, Suraikhola), Palpa (Majhuwa, Kanchakhola), Chitwan (Kuwapani, Shaktikhor) | **Trees:** *Shorea robusta, Terminala alata, Terminalia chebula, Terminalia bellerica, Adina cordifolia, Semecarpus anacardium, Terminalia chebula, Cleistocalyx operculatus, Diploknema butyracea, Lagerstroemia parviflora, Litsea monopetala, Mallotus philippensis* <br><br> **Shrubs:** *Murraya koengii, Woodfordia fruticosa, Phoenix humilis* | These forest types are common from east to west where Sal alone cannot dominate the area entirely. This type of forest shows heterogeneous species distribution representing both tall and short trees with various canopy structures. They also offer a wide array of microhabitat conditions, therefore high species diversity and productivity in these forests are seen. |
| 11 | *Schima wallichii–Shorea robusta* forest (**Chilaune-Sal ban**) (665–730 m asl) ** | Ilam, Morang | **Trees:** *Shorea robusta, Schima wallichii, Duabanga grandiflora, Terminalia alata, Lagestroemia parviflora, Syzygium cumini* **Shrubs:** *Woodfordia fruticosa, Colebrookea oppositifolia* | Disturbed forest in the Chure region of Ilam, may *Schima wallichii* soon replace *Shorea robusta* if disturbances (mostly logging) continue. |
| 12 | *Pinus roxburghii–Shorea robusta* forest (**Khote Sallo-Sal ban**) (440 m asl) *** | Bara (3 no. Khola) | **Trees:** *Shorea robusta, Pinus roxburghii, Semecarpus anacardium, Albizia lebbeck* **Shrubs:** *Woodfordia fruticosa, Inula cappa, Berberis asiatica* | These types of associations are developing in Chure, Observed in Bara, and probably occurs also in Makwanpur. |

| | | | | |
|---|---|---|---|---|
| 13 | *Pinus roxburghii* forest (**Khote sallo ban**) (435–1200 m asl) ** | Kailali (Khanidanda, Chure VDC), Kanchapur (Bedkot), Bara (3 no. Khola) Daldeldhura | **Trees:** *Pinus roxburghii, Myrica esculenta, Quercus lanata, Semecarpus anacardium, Lagestroemia parviflora* **Shrubs***: Rubus ellipticus Phoenix humilis, Berberis asiatica, Inula cappa* | Pine forests are found in Chure region of western and Central Nepal in dry north facing slopes. It reduces the growth of other native tree species to its range by forming thick mat of fallen dwarf shoots on the ground. A Forest type is characterized by continuous canopy and very poor understory. |
| 14 | Bamboo thickets (**Bans ban**) (113 m asl) * | Siraha (Baba Tal) | Monospecific thickets; no other vegetation- ground is completely covered with the litter from the bamboos and there are no other shrubs, herbs, ferns or bryophytes. | Probably colonized after disturbance (landslide, forest fire); will be persisted for an unknown period until their dieback following flowering, which occurs at unknown intervals. |

Note: Threat level: Low = *, Moderate =**, High = ***, Very High = ****. The single asterisk was given when there was no or at least one evidence of disturbance. Likewise, two asterisks were given when there was two evidence of disturbance, three asterisks for three evidence of disturbance, and four asterisks were given for more than three evidence of disturbance. The bold names within parentheses are Nepali names for particular forest type.

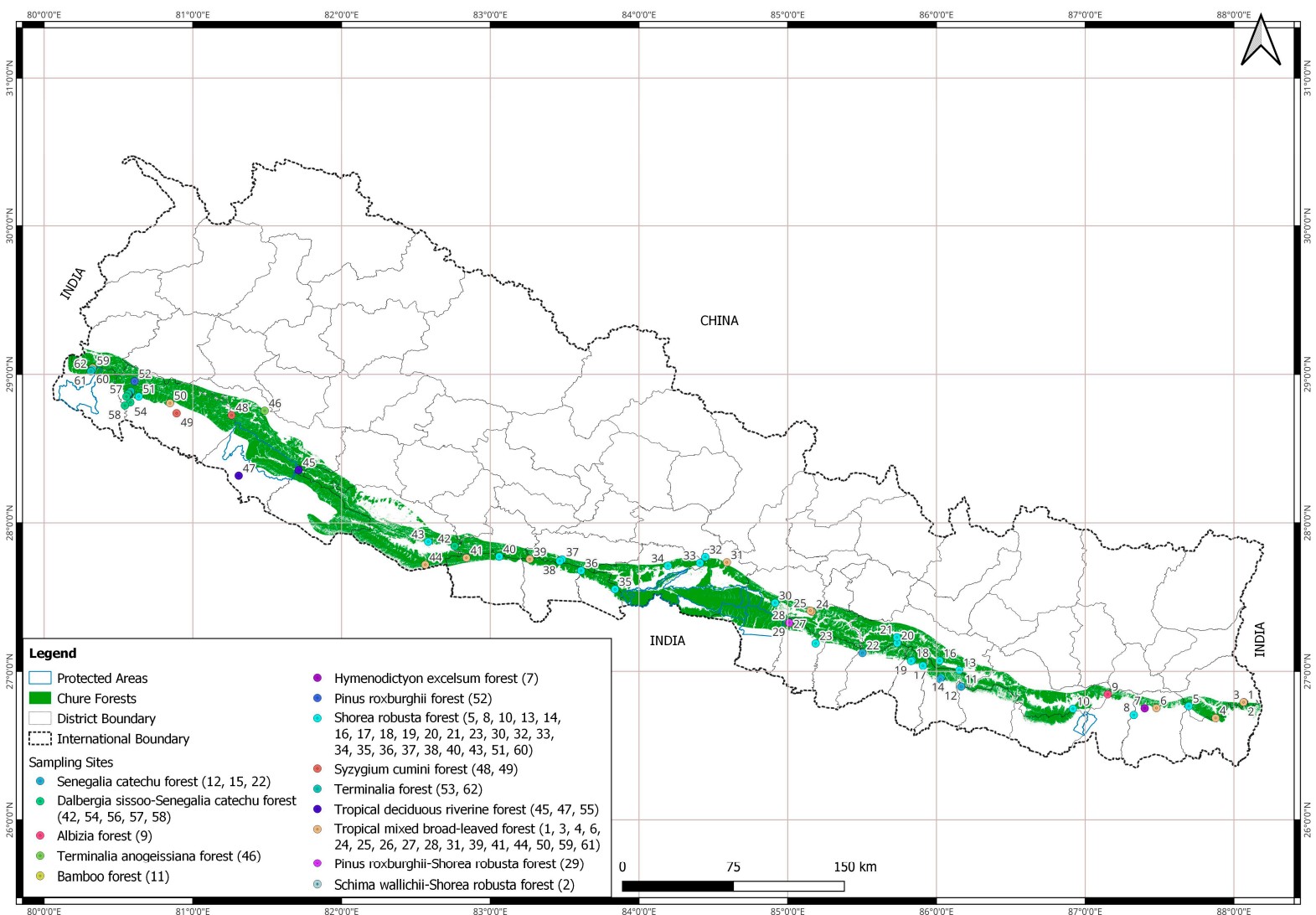

**Figure 2.** Forest ecosystem types in Chure. The numbers in the parentheses represent the sampling site(s) from where the particular ecosystem was reported. The polygons denote the protected areas.

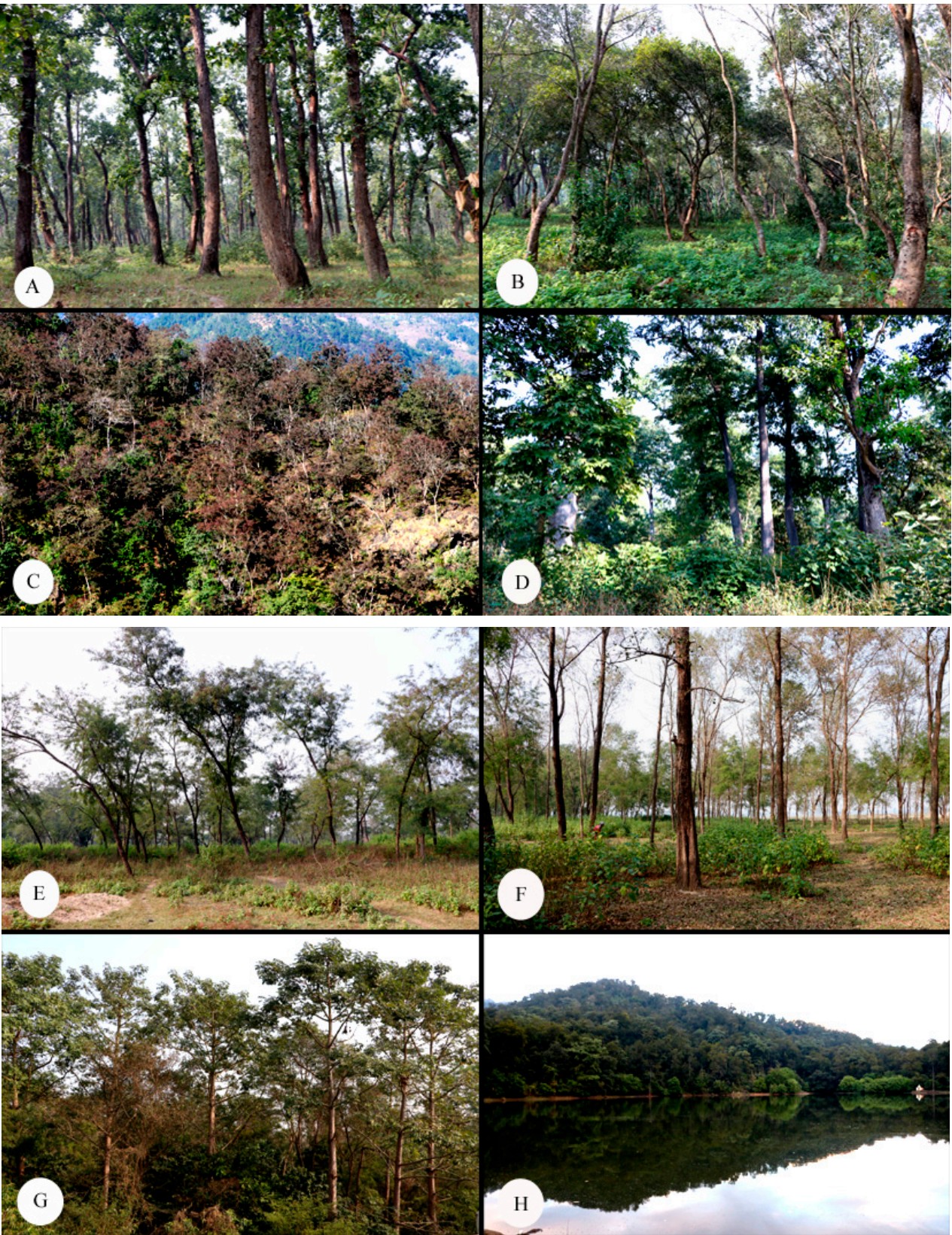

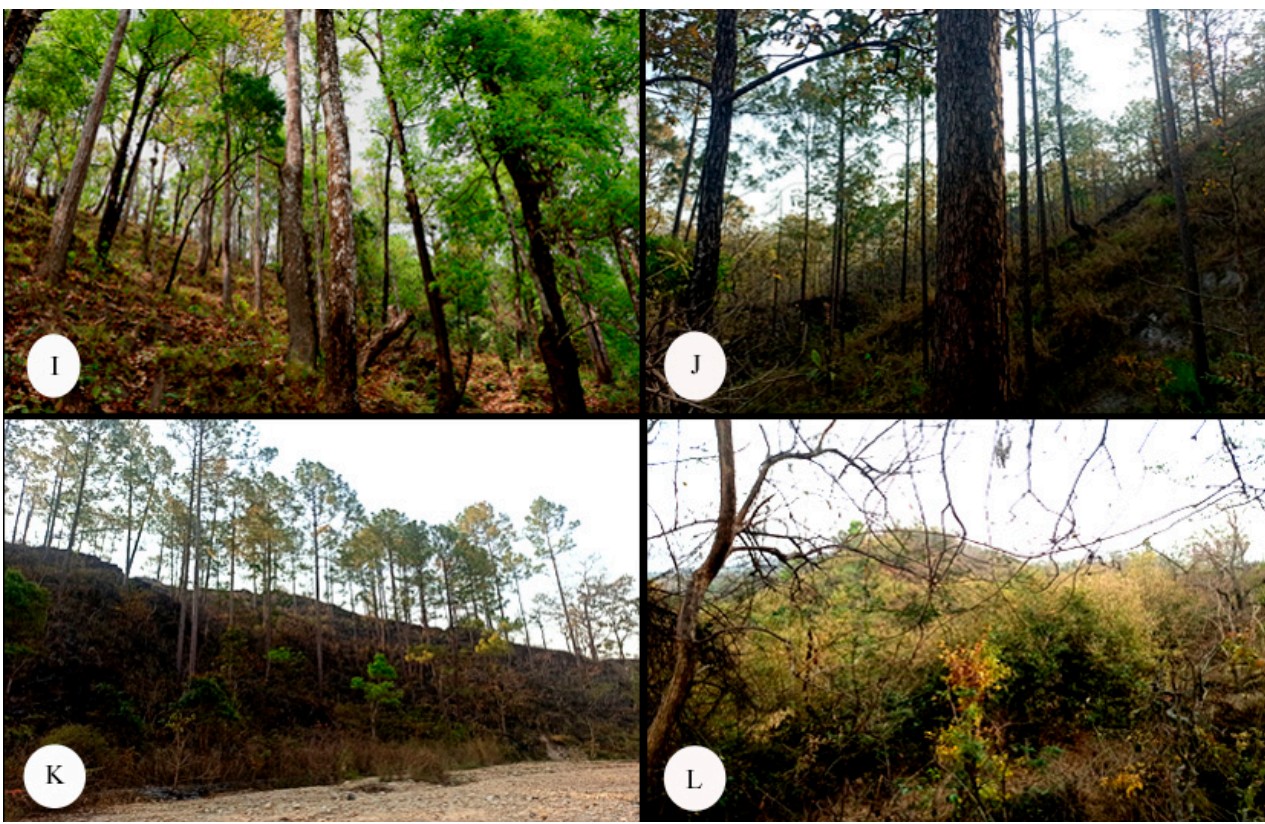

**Figure 3.** Phtoplates showing some of the forest ecosystem types in Chure. (**A**). *Shorea robusta* forest ecosystem, (**B**). *Syzygium cumini* forest ecosystem, (**C**). *Terminalia anogeissiana* forest ecosystem, (**D**). *Terminalia* forest ecosystem, (**E**). *Senegalia catechu* forest ecosystem, (**F**). *Dalbergia sissoo–Senegalia catechu* forest ecosystem, (**G**). Tropical deciduous riverine forest ecosystem, (**H**). Tropical mixed broadleaved forest ecosystem, (**I**). *Schima wallichii–Shorea robusta* forest ecosystem, (**J**). *Pinus roxburghii–Shorea robusta* forest ecosystem, (**K**). *Pinus roxburghii* forest ecosystem, (**L**). Bamboo thickets.

**Table 2.** Forest ecosystems in Chure.

| SN | Type of Ecosystem | Region | Reported in BPP Report (and Referred in Chure Master Plan) | Reported as New |
|----|-------------------|--------|----|----|
| 1 | *Shorea robusta* forest | East, Central, West | √ | |
| 2 | *Hymenodictyon excelsum* forest | East | | √ |
| 3 | *Syzygium cumini* forest | West | | √ |
| 4 | *Terminalia anogeissiana* forest | West | | √ |
| 5 | *Terminalia* forest | West | √ | |
| 6 | *Senegalia catechu* forest | East | √ | |
| 7 | *Albizia* forest | East | √ | |
| 8 | *Dalbergia sissoo–Senegalia catechu* forest | West | √ | |
| 9 | Tropical deciduous riverine forest | East and West | √ | |
| 10 | Tropical mixed broadleaved forest | East, Central, West | √ | |
| 11 | *Schima wallichii–Shorea robusta* forest | East | | √ |
| 12 | *Pinus roxburghii–Shorea robusta* forest | Central | | √ |
| 13 | *Pinus roxburghii* forest | Central, West | | √ |
| 14 | Bamboo thickets | East | | √ |

*3.2. Threats and Vulnerabilities*

Of the total 62 sampling sites, forest encroachment and deforestation were found in 56 sites (90% plots) followed by forest fire and invasive species in 16 sites each (26% plots). Further, based on the detailed assessment of disturbance variables, we identified one ecosystem as very highly threatened (with four asterisk marks—*Senegalia catechu* Forest), four ecosystems as highly threatened (with three asterisk marks—*Terminalia* Forest, *Dalbergia sissoo–Senegalia catechu* Forest, Tropical mixed broadleaved forest, *Pinus roxburghii–Shorea robusta* Forest), six as moderately threatened (with two asterisk marks—*Shorea robusta* Forest, *Hymenodictyon excelsum* Forest, *Albizia* Forest, Tropical deciduous riverine Forest, *Schima wallichii –Shorea robusta* Forest, and *Pinus roxburghii* Forest), and three as relatively less disturbed (with one asterisk mark—*Syzygium cumini* Forest, *Terminalia anogeissiana* Forest, and Bamboo thickets) (see asterisk marks in Table 1). Forest disturbances (grazing, logging, fire, flood cutting, encroachment, fuelwood collection) were high in *Terminalia* Forest, *Senegalia catechu* Forest, *Dalbergia sissoo–Senegalia catechu* Forest, Tropical mixed broadleaved Forest, and *Pinus roxburghii–Shorea robusta* Forest, moderate in *Shorea robusta* Forests, *Hymenodictyon excelsum* Forest, *Albizia* Forest, Tropical deciduous riverine Forest, *Schima wallichii–Shorea robusta* Forest, and *Pinus roxburghii* Forest, whereas low in *Syzygium cumini* Forest, *Terminalia anogeissiana* Forest, and Bamboo thickets (Figure 3). It was found that the existence of the *Senegalia catechu* forest is limited in very few localities of Chure as fragmented patches. These forests are one of the highly threatened forest ecosystems in the Chure region mainly due to forest encroachment, deforestation, road construction, and invasion of other species.

Forest ecosystems plagued with the high level of threat caused by disturbances such as logging (mostly illegal) and poor regeneration and survival (in the case of *Senegalia catechu*, *Dalbergia sissoo–Senegalia catechu*) are more vulnerable. The mature individuals of *Senegalia catechu* in the Chure forest are extremely rare, mostly due to the illegal felling. The conservation priorities should be focused on the protection of this threatened species along with other associated species such as *Adina cordifolia* and *Shorea robusta*.

*3.3. Forest Regeneration*

Out of 14 forest ecosystem types reported from Chure, six forest types namely *Shorea robusta*, *Hymenodictyon excelsum*, *Syzygium cumini*, Tropical mixed broadleaved forests, *Schima wallichii–Shorea robusta*, and Bamboo thickets are producing a good number of seedlings (>60 individuals per plot) and naturally regenerating well. Adequate regeneration of these species was also reported by DFRS [16]. This might be due to the fact that these forests are relatively less disturbed. Similarly, four forest ecosystems showed moderate germination showing less than 60 and more than 20 individuals within the plot studied. Four forest types namely *Terminalia anogeissiana*, *Senegalia catechu*, *Dalbergia sissoo–Senegalia catechu*, and *Albizia* forests showed poor natural regeneration representing less number of seedlings in the plot (Figure 4). The poor representation of seedlings in the plot of these forest types might be due to high disturbances like grazing, deforestation, fire, flood, and invasive species.

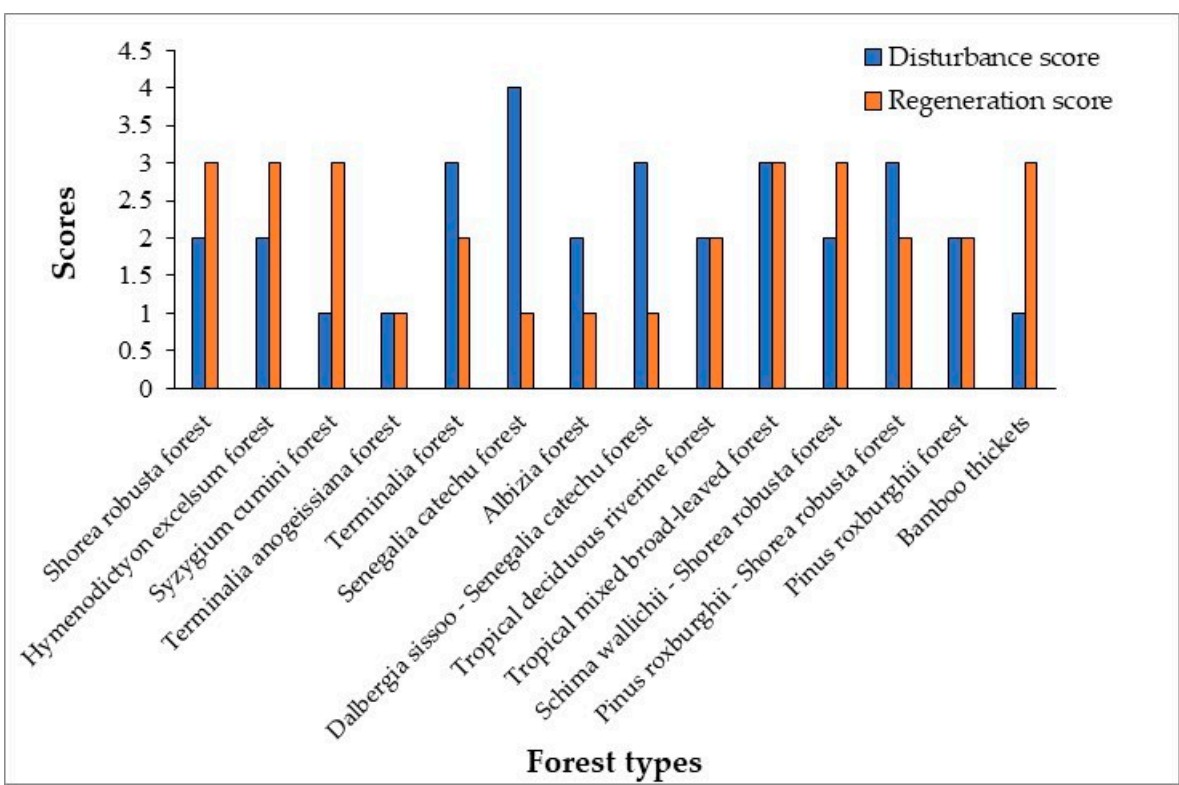

**Figure 4.** Disturbance and regeneration score according to forest ecosystem types in Chure landscape.

The forest regeneration status is satisfactory in Chure as most of the forests showed moderate to high regeneration. This is because seventy-two percent of our sampling plots were located in community forests meaning that the user groups put some restrictions on resource extraction. However, still, some of the forests showed poor regeneration. Moreover, our sampling design purposively selected relatively good-quality less disturbed forests. Therefore, the overall forest quality of Chure could be much lower than reported in our study (see Figure 5a).

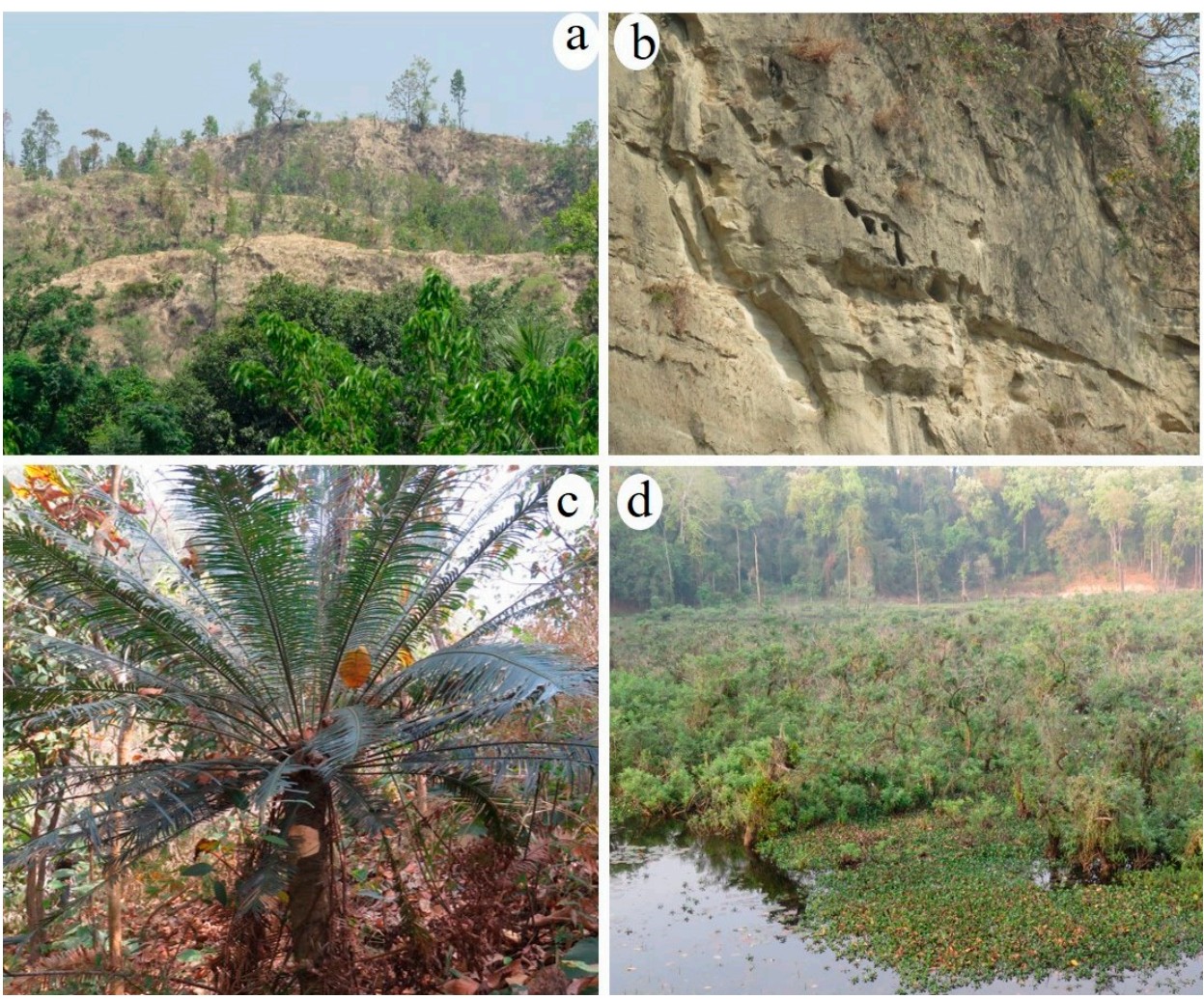

**Figure 5.** One of the most degraded sites in Chure (**a**), young and fragile deposits in Chure as nesting sites (**b**), endangered *Cycas pectinata* (**c**), freshwater swamp forests of *Cephalanthus tetrandra* (**d**).

*3.4. Floral and Faunal Biodiversity of Chure*

The Chure harbors about 1000 plant species of different life forms [16,29,47]. The floral diversity is comparable with that of the Indian part of Chure [54]. It includes more than 281 tree species that constitute about 40% of total tree species recorded in Nepal, 186 shrubs, and 322 herbs including Pteridophytes [16]. Of the total 293 endemic plants of Nepal [55], some nine species are found in the Chure region including *Begonia minicarpa* H. Hara (Locality-Sunsari, 630 m); *B. tribenensis* C.R. Rao (Sunsari, 130 m); *Eriocaulonex sertum* Satake (Jhapa, 200–300 m); *E. obclavatum* Satake (Jhapa, 200–300 m); *Isodondhan kutanus* Murata (Dhankuta, 1200 m); *Eria nepalensis* D.M. Bajracharya & K.K. Shrestha (Chitwan, 200 m); *Malaxis tamurensis* Tuyama (Dhankuta, 1200 m); *Ophiorrhiza nepalensis* Deb & Mondal (Ilam, 450 m); *Salix plectiles* Kimura (E. Nepal, 200 m) [29,55,56]. However, a detailed floristic inventory of the Chure is lacking.

Chure is also a pristine habitat for at least 41 species of mammals, 99 species of herpetofauna (24 species of amphibians and 75 species of reptiles), and 279 species of butterflies [17,19]. Most of the species found in the region are also listed in the CITES Appendices and IUCN Red data book, e.g., tiger, elephant, and rhinoceros. Megafauna like tiger, elephant, rhinoceros, wild buffalo, bison, and many others are flagship species to the Chure and lowland ecosystems. As the landscape is rich in biodiversity, the Government of Nepal has established networks of protected areas in Terai-Chure to conserve mostly the major faunal species. Of these protected areas, Chitwan National Park, Parsa National

Park, Bardiya National Park, and Banke National Park cover parts of Chure region. Likewise, four Ramsar sites in region namely Koshi Tappu, Beesshhajari and Associated Lakes, Jagadishpur Reservoir, and Ghodaghodi Lake provide important habitat for wildlife. Ramdhuni (Sunsari), Rauta (Udaypur), Dhanushadham (Dhanusha), Barandabhar (Chitwan), Khata (Bardiya), Kakrebihar (Surkhet), Basanta (Kailai) and Laljhadi-Mohana (Kanchanpur) are protection forests in Chure-Terai [34]. Young and fragile deposits in Chure provide important nesting sites for birds (Figure 5b).

*3.5. Biodiversity Hotspots*

The whole range of Chure landscape in Nepal from the east to the west is a biodiversity hotspot [29]. Of the total 27 Important Bird Areas of Nepal, 13 are located in Chure-Terai [57]. They are Barandabhar Forest and Wetlands, Bardia National Park, Chitwan National Park; Dang Deukhuri Foothill Forests and West Rapti Wetlands, Dharan Forests, Ghodaghodi Lake, Jagadishpur Reservoir, Koshi Tappu Wildlife Reserve and Barrage, Farmlands in Lumbini, Nawalparasi Forests, Parsa Wildlife Reserve, Shukla Phanta Wildlife Reserve, and Urlabari Forest Groves. Likewise, Chure forests are important habitats for species under different conservation status. We recorded the endangered *Cycas pectinata* species (a member of a group called Cycads which are an ancient group of seed plants that originated over 280 million years ago [58] in its natural habitats at central (Makawanpur) and eastern Nepal (Andha Rajarani area of Ilam, Bhatighari Communnity Forest (CF) of Danusha) (Figure 5c). Likewise, Bashyal et al. [59] reported several important sites for *Cycas pectinata* in central Nepal. Bhuju and Joshi [27] identified important sites for tree fern (*Cyathea spinulosa)* from eastern Nepal (Madi in Morang and Bajho and Mahmai in Ilam and Bajhoand Chisapani in Ilam). Likewise, Chure forests in far-west Nepal constitute a good population of mature *Pterocarpus marsupium* and *Dalbergia latifolia* forest (Bedkot Tal, Kanchanpur). There are also good natural forest patches of *Senegalia catechu*, a threatened species**,** in Chure forests of eastern Nepal **(**Sarlahi-Lopchan Tol, Dhanusha-Bhatighari CF Puspabanpur, Siraha-Baba Tal, Bandipur- 3) (Table 1). These primary forests are important from conservation point of view and the areas can be termed Important Plant Areas (IPAs). We propose three additional IPAs in Chure landscape to the list of Hamilton and Radford [60] (Table 3). Apart from protected areas and Ramsar sites in Chure-Terai, the Dang-Deukhuri Foothill Forests and West Rapti Wetlands, Dharan Forests, Nawalparasi forests, Farmlands in Lumbini and Urlabari Forest Groves are Important Bird Areas [57].

**Table 3.** Important Plant Areas in Chure.

| IPA Complex | Number of Site | District(s) |
|---|---|---|
| Lower Mahakali–Seti | 1 + 1 | Dadeldhura, Kanchanpur |
| Lower Bheri–Rapti | 2 | Salyan and Surkhet |
| Terai Arc Landscape | 8 | Kailali, Bardia, Banke, Dang, Palpa, Nawalparasi, Chitwan, Parsa |
| Rapti–Lumbini | 2 | Pyuthan and Argahkhanchi |
| Narayani | 2 | Makwanpur and Bara |
| Lower Janakpur | 2 + 1 | Sarlahi, Sindhuli, Dhanusa |
| Udayapur | 1 | Udayapur |
| Morang | 1 | Morang |
| Lower Kangchenjungha | 1 + 1 | Ilam, Jhapa |

Modified from Hamilton and Radford [60]. Added one site each in Kanchanpur, Dhanusa and also suggested one site in Morang.

We explored some of the important wetlands of Chure landscape during the field study (Chuli Pokhari, Ilam; Rajarani Tal, Morang; Baba Tal, Siraha; and Bedkot Tal, Kanchanpur). These wetlands and catchment areas are exceptionally rich in biodiversity. For example, we found good population of *Cephalanthus tetrandra* in the natural habitat in the

Rajarani Tal of Morang district within Chure landscape (Figure 5d). The freshwater swamp forests of *Cephalanthus tetrandra* trees are believed to be rare in South East Asia. Mikhama and Sirisant [61] have reported *C. tetrandra* freshwater swamp forests from Don Daeng village, Nakhon Phanom province as the only one of its kind in Northeast Thailand, and they have highlighted the ecological role of *C. tetrandra* forest and the active involvement of local people for managing and conserving the important wetland tree species. Furthermore, wetlands play important role in maintaining hydro-climatic balance, and control of flood and landslide both upstream and downstream along the Chure landscape. Further inventory of wetlands in Chure would provide important information for identifying biodiversity hotspots and designing biodiversity conservation programs accordingly.

The human-dominated landscapes within Chure such as in Makwanpur and Chitwan which are inhabited by the Chepang ethnic group are also important for the preservation of biocultural diversity as these landscapes hold Chiuri (*Diploknema butyracea*), a cultural keystone species for Chepang [62].

### 3.6. Threatened Plant Species

Forest ecosystems in Chure provide important habitats for twenty-six plant species with different conservation status. This further signifies the conservation importance of Chure from the biodiversity point of view (Table 4). Of 26 species documented, five species are Endangered (IUCN); four species each fall under the protected plant list of the Government of Nepal, CITES appendix II; and Vulnerable category of CAMP; three species are in IUCN rare category, two species are in Endangered category of CAMP and one species each is in Threatened (IUCN), Commercially threatened (IUCN) and CITES III list.

**Table 4.** Plant species with different conservation status in Chure.

| SN | Species | Category |
|---|---|---|
| 1 | *Alstonia scholaris* (L.) R.Br. | Rare (IUCN category) |
| 2 | *Asparagus racemosus* Willd. | Vulnerable (Conservation Assessment and Management Planning, CAMP *) |
| 3 | *Bombax ceiba* L. | Nationally protected (Under the National list of timber trees banned for felling, transportation or export) |
| 4 | *Butea monosperma* (Lam.) Kuntze | Endangered (IUCN category) |
| 5 | *Choerospondias axillaris* (Roxb.) B.L. Burtt & A.W. Hill | Rare (IUCN category) |
| 6 | *Cinnamomum glaucescens* (Nees.) B.L.Burtt & A.W.Hill | Protected |
| 7 | *Crateva unilocularis* Buch.-Ham. | Rare |
| 8 | *Curculigo orchioides* Gaertn. | Vulnerable (CAMP) |
| 9 | *Cycas pectinata* Griff. | CITES Appendix II; Endangered (IUCN category) |
| 10 | *Dalbergia latifolia* Roxb. | Nationally protected (Under the National list of timber trees banned for felling, transportation or export) |
| 11 | *Dendrobium fimbriatum* Hook. | CITES Appendix II |
| 12 | *Dioscorea deltoidea* Wall. Ex Griseb. | Commercially threatened |
| 13 | *Magnolia champaca* (L.) Baill. Ex Pierre | Endangered (IUCN category) |
| 14 | *Gnetum montanum* Markgr. | CITES Appendix III; Endangered (IUCN category) |
| 15 | *Operculina turpethum* (L.) Silva Manso | Endangered (CAMP) |
| 16 | *Oroxylum indicum* (L.) Kurz | Vulnerable (IUCN category) |
| 17 | *Piper longum* L. | Vulnerable (CAMP) |
| 18 | *Pterocarpus marsupium* Roxb. | Nationally protected (Under the National list of timber trees banned for felling, transportation or export) |

| 19 | *Rauvolfia serpentina* (L.) Benth. Ex Kurz | Endangered (IUCN category) |
| 20 | *Rhynchostylis retusa* (L.) Blume | CITES Appendix II |
| 21 | *Rubia manjith* Roxb. Ex Fleming | Vulnerable (CAMP) |
| 22 | *Senegalia catechu* (L.f.) Willd. | Threatened (IUCN category) |
| 23 | *Shorea robusta* Gaertn. | Nationally protected (Under the National list of timber trees banned for felling, transportation or export) |
| 24 | *Swertia angustifolia* Buch.-Ham. Ex D.Don | Endangered (CAMP) |
| 25 | *Tinospora sinensis* (Lour.) Merr. | Vulnerable (CAMP) |
| 26 | *Vanda tessellata* (Roxb.) Hook. ex G.Don | CITES Appendix II |

* Bhattarai et al. [63].

## 4. Conclusions and Recommendations

The forest ecosystems in Chure are diverse and dynamic. We revisited the forest ecosystems outside protected areas in Chure and found that 14 forest ecosystem types are available in the Chure landscape of Nepal. The reference ecosystem types in Chure were taken from the Biodiversity Profiles Project [15] where 11 forest ecosystems are reported to occur outside protected areas in Chure. Our study further shows that the BPP reported *Alnus nitida* riverine forest in the west does not occur in Chure. The present study reported *Hymenodictyon excelsum* Forest, *Syzygium cumini* Forest, *Terminalia anogeissiana* Forest, *Schima wallichii–Shorea robusta* Forest, *Pinus roxburghii–Shorea robusta* Forest, *Pinus roxburghii* Forest, and Bamboo thickets as new forest ecosystems in Chure because they are not reported previously. These newly formed associations (forest types) are at the preliminary stages of forest ecosystem development, as these types are not common in east-west of Chure landscapes, but slowly evolving. As some of the forest ecosystems such as *Hymenodictyon excelsum* Forest and *Dalbergia sissoo–Senegalia catechu* Forest are found in Terai, mostly on Chure flood plains, these types are included in the total count as they form the contiguous ecosystems in Terai-Chure.

Though our study only located the forest ecosystems, we emphasize that this study complements what is existed for Chure at present and the ongoing national-level ecosystem mapping project of the Ministry of Forests and Environment. The present findings are also helpful to design the conservation programs targeting forest ecosystems in Chure. We affirm with caution that our study is comprehensive so far, but we do not claim that the study is complete as many remote and inaccessible areas were not covered during the field survey. Based on our study, we recommend some priority activities for biodiversity conservation and ecosystem restoration to be implemented in the Chure landscape as below.

1. Our study has identified biodiversity hotspots based on species richness and the occurrence of different plant species under different conservation categories. These areas should be given priority for conservation. Special status could be suggested for the conservation of such species, for example, the Andha Rajarani area in Ilam can be given the status of the park or botanical garden as it is one of the diverse areas for biodiversity.

2. Some of the hotspots identified in this survey are in the community forests, so the local people can be informed about the conservation values of these ecosystems and they can make aware and educated. The CFs operational plan could be revised to incorporate the conservation values of these ecosystems.

3. As ecosystems are dynamic in nature their existence could be of short-term as well. Therefore, long-term monitoring of vulnerable ecosystems such as *Senegalia catechu, Dalbergia sissoo–Senegalia catechu,* and Bamboo ecosystems is important. Rapidly changing land use patterns and climate change may put additional pressure on Chure ecosystems.

4. Restoration of the degraded ecosystems such as *Shorea robusta* Forest Ecosystems, *Senegalia catechu* Forest Ecosystems, and *Dalbergia sissoo–Senegalia catechu* Forest Ecosystems should be given high priority. At present, the restoration activities are not ecosystem focused and there is no active participation of the communities. Local governments should be informed about the ecosystem types and their conservation values within their jurisdiction.

**Supplementary Materials:** The following supporting information can be downloaded at: https://www.mdpi.com/article/10.3390/f14010100/s1, Table S1: Location of the forest ecosystem type and GPS coordinates, Table S2: Forest ecosystems in Chure and Terai**.**

**Author Contributions:** Conceptualization, Y.U., A.T., S.K., R.K.P.Y., S.S., S.G., K.P. and M.D.; methodology, Y.U., A.T., S.K. and A.C.; formal analysis, Y.U., A.T., S.K. and A.C.; writing—original draft preparation, Y.U., A.T. and S.K.; funding acquisition, Y.U. All authors have read and agreed to the published version of the manuscript.

**Funding:** This research was funded by the President Chure Terai-Madhesh Conservation Development Board, Nepal. Article publication fee was supported by the University Grants Commission Nepal.

**Data Availability Statement:** The data included in this study are available upon request from the corresponding authors.

**Acknowledgments:** We are thankful to Pramod Kumar Jha, Krishna Prasad Oli, and Hemlal Aryal for constructive suggestions to improve the manuscript.

**Conflicts of Interest:** The authors declare no conflict of interest.

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
