# Peer review of "Characterization of Forest Ecosystems in the Chure (Siwalik Hills) Landscape of Nepal Himalaya and Their Conservation Need"

_forests, doi:10.3390/f14010100_

Round 1

Reviewer 1 Report

In this paper, vegetation in the Chure mountains of Nepal was investigated through extensive sampling (62 sample sites) and 14 forest types were summarized. In Nepal, there is a gap in this type of study, and this study can fill that gap well. It provides important basic information for our understanding of forest communities in the Himalayas, a world biodiversity hotspot.

For this reason, we need to conduct qualitative and quantitative research in a scientific way in accordance with certain modern ecological standards. I think the author's manuscript also reflects some shortcomings in this regard, as follows:

First of all, the author is unclear about the concepts of Vegetation type, Forest Type and Ecosystem type, and uses confusion in the text. Based on the results shown by the author in the article (the composition of the taxa of trees and shrubs), I think it is appropriate to call it Forest Type. Because the other two need to be described in greater detail for herbs, epiphytes, mosses, and even animals.

The author's classification of Forest Type is not standardized, there is no systematic division according to the hierarchy, and the naming of some types is not very rigorous. For example, among the 14 vegetation types, except for Pinus roxburghii forest and Bamboo thickets, they all belong to the Tropical monsoon forest category, and the 12 types under them can only be called "Subtype", in addition, I suggest that the arrangement of types in Table 1 should also be systematic, such as listing the types dominated by Shorea next to each other. Fabaceae-based types are listed next to each other, etc.

"Characterization" is not only a qualitative and simple description of forest communities, but also requires quantitative and standard community research. The author sets up a large number of quadrats, but from the "Methods" paragraph, it seems that only the coverage of each tree species is measured. Did the authors make other standard community measurements in the plot, such as DBH, crown amplitude, number, etc.? If so, I recommend listing quadrat statistics for major vegetation types in this article, such as Relative frequency, Abundance, Relative dominace, The importance value index (IVI) and etc. This is the most basic information that community studies need to provide, rather than simply listing the main types of species composition in Table 1.

I strongly recommend adding plates for 14 forest type appearance photos. As a basic information, photos of the appearance of the community are indispensable. The journal has no control over the number of plates.

The evaluation system of regeneration status and disturbances was designed by the author himself? If not, please indicate the basis for its reference in the text?

Figure 3 is completely unnecessary, just describe it in the paragraph. The histogram of Figure 4&5 is not necessary, this is not doing PPT, the information of the two figures can be combined into a single table.

Is Table 4 completely excerpted from Bhattarai et al.? If so, there is no need to list it in this article at all.

Reviewer 2 Report

The manuscript by Uprety et al. focuses in describing forest types located outside natural protected areas from the Chure region, Nepal. Although I appreciate the effort of the authors for collecting field data, the scope of the study is local and the manuscript lack of a solid scientific background. The terminology used by the authors for defining “forest ecosystems” is quite confusing, as the spatial scale they used is extremely small (0.5 ha). The methods must be largely improved to make them replicable, mainly because the criteria used by the authors for establishing threat levels of the forest seem arbitrary, while they should be founded on a deep scientific reasoning. The results and discussions section includes a lot of speculative information that is neither aligned with the findings nor the aims of the study and, therefore, it is hard to understand. I recommend to the authors to make and in-deep revision of the literature and rewrite the manuscript in the light of the state-of-the-art in forest conservation, rather than providing a description of a focal ecosystem. I am sure they have the data for this, and perhaps the manuscript should be focused on the methods they propose for assessing the threat levels fo forest in the context of landscape ecology.

General comment on the introduction: The introduction of the manuscript is fine, but it almost fully focuses in describing the Chure mountain range. It would be important to provide more details about the novelty and relevance of this study in the context of forest ecology and conservation to state a solid theoretical framework for supporting the aims of the study. As currently presented, the manuscript just seems like a descriptive study.

Methods, Figure 1: In this figure, the authors show the location of the sampling points. In the subsection “Sampling procedures”, however, they indicate that only forest ecosystems located outside protected areas were sampled. For this reason, I recommend to include the polygons of the protected areas in this map (Figure 1) to facilitate the understanding of the procedure used to select the sampling sites.

Methods, lines 125-126: I am not sure that the criteria used by the authors to delimit a forest ecosystem is useful. A forest covering a minimum area of 0.5 ha cannot be considered a distinct ecosystem, but it could be considered a habitat within a forest landscape. In this way, a landscape or ecosystem would be composed by several habitat types that spatiotemporally interact among them. For this reason, I encourage to the authors to revise the concepts of ecosystem, landscape and habitat and clearly define the spatial scale (and concepts) they are using in this study. This could also contribute to establish a more solid theoretical framework for the study.

Methods, line 129: How did the authors establish the conservation status of each forest? What criteria did they use for determining this? They must describe these criteria in the methods to make the study replicable.

Methods, line 137: Why field sampling was conducted from November 2021 to February 2022? The authors should indicate if they had some special reason for sampling the forest in this particular season of the year (e.g., trees were flowering and this facilitates the identification of species, or any other reason).

Methods, lines 143-145: What ecological/conservation principle was used to establish the regeneration status of forests and their degree of disturbance? There is some theoretical/empirical reason for using these criteria? The authors must justify the criteria they used for assessing the regeneration status and degree of disturbance of forests. Otherwise, the readers can assume that these are arbitrary criteria with no solid foundations.

Methods, lines 153-156: The authors must move this “asterisk method” to the caption of Table 1, rather than describing this in the methods. In the methods, instead, the authors should provide a better description of the procedures they used to determine the threat level of forests.

Results and Discussion, line 179: Replace “including the Bamboo thickets in the Siraha district of east Nepal” with “,including the Bamboo thickets in the Siraha district of east Nepal,” (note the use of the commas for separating this sentence from the resto of the text).

Results and Discussion, line 180: Replace “BPP” with “BPP reports”.

Results and Discussion, lines 181-184: Once again, I think that the authors are identifying “tree stand types” (i.e., tree stand dominated by some species) rather than “ecosystem types”. They must revise the terminology used for describing the results to make them clearer.

Results and Discussion, Figure 2: The authors must increase color contrast of the dots used for denoting the forest types to facilitate their visualization in the map. Indeed, larger dots would be also useful.

Results and Discussion, lines 186-196: Why this section focuses in describing forest stands dominated by Alnus nitida? The authors indicate that they did not find stands dominated by this tree species, but they focus this section in explain the absence of this forest type in the studied region. This makes no sense.

Results and Discussion, lines 208-210 and Figure 3: Figure 3 must be removed from the manuscript, as all information of this bar chart is provided din the text (the authors should avoid redundancy between text, figures and tables).

Results and Discussion, lines 231-238: Revise the use of italics in the scientific name of species.

Results and Discussion, lines 243-245: Why did the authors make this recommendation? Are these species unprotected? Better justifications are required to support the recommendation.

Results and Discussion, Figure 5: This figure can be combined with figure 4, as both figures provide information on the status of forest types, and they refer to scores that the authors assigned. Pepahs a dual bar chart would be useful.  

Results and Discussion, lines 273-365: These three sections are merely descriptive and they are not related with the findings of the study. Thus, I cannot understand the reason of providing these descriptions in the context of the study and its aims. All these sections sections must be rewritten in accordance with the results of the study.

Round 2

Reviewer 1 Report

I'm glad you were able to complete the changes quickly as requested. Now with only a few minor changes, it can be published: the order of the different communities in Table 1 still seems a little strange, I suggest you arrange them in the order of "monodominant broadleaf forest, mixed broadleaf forest, pine forest, bamboo forest", and the monodominant broadleaf forest in the order of Shorea, Terminalia, Fabaceae, and others.

Author Response

Thank you for this comment. We have rearranged Table 1 as suggested and made the subsequent changes in the abstract, Table 2, and text. 

Reviewer 2 Report

The authors addressed most the comments I made and I am satisfied with the answers they provide. Thus, I feel that the manuscript could be published.

Author Response

Thank you so much for your very constructive comments and for supporting us to improve the manuscript significantly.